# A Novel Localization Technology Based on DV-Hop for Future Internet of Things

Xiaoying Yang [1,2], Wanli Zhang [1,3,*], Chengfang Tan [1,2] and Tongqing Liao [2]

1   School of Information Engineering, Suzhou University, Suzhou 234000, China;
    yangxiaoying@ahszu.edu.cn (X.Y.)
2   School of Electronic Information Engineering, Anhui University, Hefei 230039, China
3   Anhui Provincial Key Laboratory of Intelligent Building and Building Energy Conservation, Anhui Jianzhu
    University, Hefei 230022, China
*   Correspondence: zhangwanli@ahszu.edu.cn; Tel.: +86-181-5577-7889

**Abstract:** In recent years, localization has become a hot issue in many applications of the Internet of Things (*IoT*). The distance vector-hop (*DV-Hop*) algorithm is accepted for many fields due to its uncomplicated, low-budget, and common hardware, but it has the disadvantage of low positioning accuracy. To solve this issue, an improved *DV-Hop* algorithm—*TWGDV-Hop*—is put forward in this article. Firstly, the position is broadcast by using three communication radii, the hop is subdivided, and a hop difference correction coefficient is introduced to correct hops between nodes to make them more accurate. Then, the strategy of the square error fitness function is spent in calculating the average distance per hop (*ADPH*), and the distance weighting factor is added to jointly modify *ADPH* to make them more accurate. Finally, a good point set and *Levy* flight strategy both are introduced into gray wolf algorithm (*GWO*) to enhance ergodic property and capacity for unfettering the local optimum of it. Then, the improved *GWO* is used to evolve the place of each node to be located, further improving the location accuracy of the node to be located. The results of simulation make known that the presented positioning algorithm has improved positioning accuracy by 51.5%, 40.35%, and 66.8% compared to original *DV-Hop* in square, X-shaped, and O-shaped random distribution environments, respectively, with time complexity somewhat increased.

**Keywords:** Internet of Things; DV-Hop; multi-communication radius; distance-weighted; grey wolf algorithm; levy flight strategy





## 1. Introduction

*IoT* refers to a network that connects any item to the Internet through information sensing devices, under agreed protocols, for autonomous information exchange, and provides useful information for people to achieve intelligent identification, positioning, tracking, monitoring, and management of things [1,2]. *IoT* is considered the world's next wave of information technology and new economic engine, which will have an immensely significant impact on the world economy, politics, culture, military, and society. *IoT* provides the potential for the development and design of a large number of applications. Currently, only a few applications have been promoted in our production and daily life. At present, most applications only have relatively primitive intelligence, but in the future, there will be many applications based on the *IoT* to improve our quality of life and production environment. The intelligent objects in these applications can communicate with each other, perceive information from the surrounding environment, and be deployed in various environments, including transportation and logistics, healthcare, production and life, and personal social media [3–6], as shown in Figure 1.

There are various aspects of *IoT* technology that are worth studying, and node localization is one of them. Accurate effective node positioning is the foundation for providing various monitoring location information services [7]. In addition, location information

can also improve the efficiency of routing in the network and achieve network topology self-configuration [8,9]. Figure 2 shows the importance of localization in many applications.

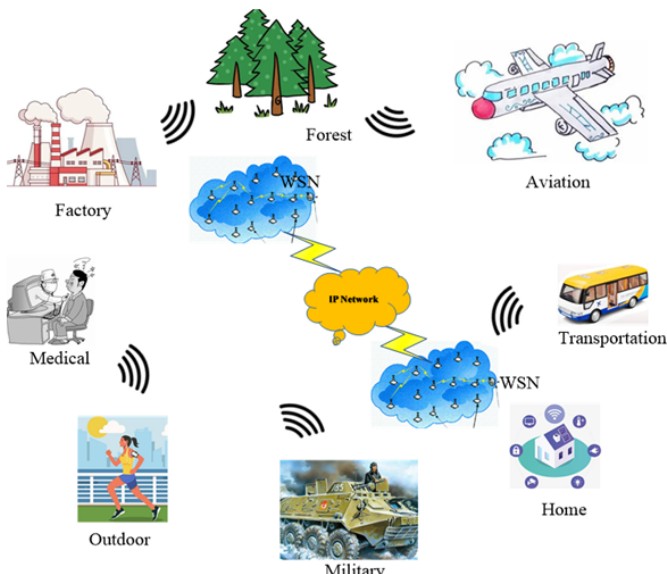

**Figure 1.** Application of *IoT*.

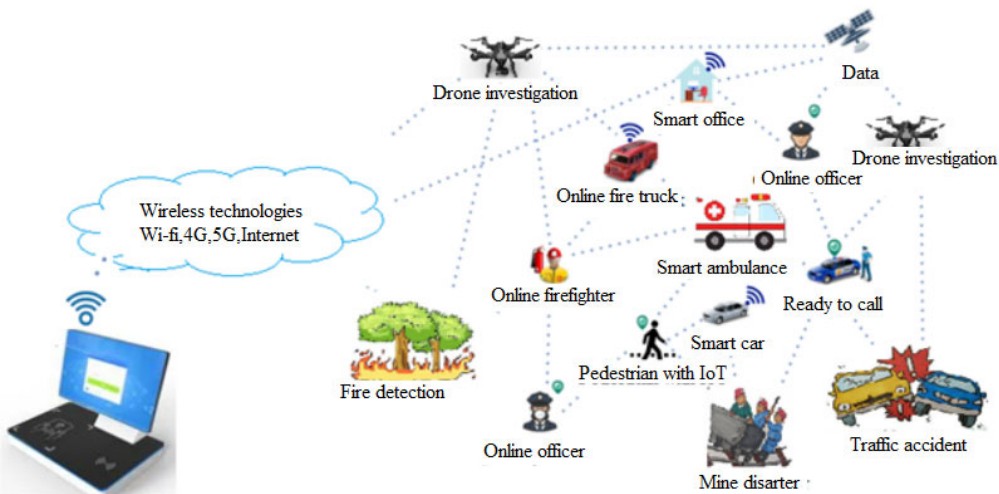

**Figure 2.** Localization in Various Applications.

According to Figure 2, a key aspect in *IoT* applications is determining the true location of devices. GPS determines the position of nodes mainly by using *GPS* receivers installed on the nodes to receive signals from multiple satellites, but this method is too expensive, bulky, and feasible. Therefore, many researchers adopt positioning methods that use known nodes (anchor nodes) to take stock of the place of unknown nodes. How to estimate the coordinates of unknown nodes through efficient and low-cost positioning algorithms has become a research focus. Su discussed various underwater localization algorithms [10]. Ullah proposed an EKF-based localization algorithm by edge computing, and a mobile robot is used to update its location concerning landmarks [11].

The most common localization classification method is chopped into ranging and ranging-free localization algorithms [12]. The former uses external hardware devices to obtain distance information between nodes, which has high localization accuracy. Common algorithms include Time of Arrival [13,14], Time Difference of Arrival [15,16], Received Signal Strength Intensity [17–19], Angle of Arrival [20,21], and so on. The latter determines the position of unknown nodes through anchor nodes, which has relatively high positioning

error but is more economical and energy-saving. There are commonly used algorithms such as Proximate PITTest [22], centroid localization algorithm [23], *DV-Hop* [24–26], etc. In applications that do not require high localization accuracy, ranging-free algorithms are more popular.

As a typical representative of nonranging algorithms, the *DV-Hop* algorithm has become a research hotspot in the engineering and academic fields. This algorithm is less affected by the environment and has low hardware requirements, which makes it for WSN with low cost, broadscale and simple node configuration [27]. Nevertheless, its localization accuracy is low. To overcome the problem, a new improved *DV-Hop* called *DV-Hop* is put forward in this paper. The contributions of this paper are:

(1)  Use three communication radii to broadcast messages and decimate the hops to lessen errors created by varying hop lengths. Calculate the distance between nodes within onehop using the virtual intersecting circle geometry method. Then, a jump difference correction coefficient is introduced to further correct the minimum number of hops;

(2)  *ADPH* of anchor node is calculated by using the square criterion to minimize various errors and introduce a distance weighting factor to cut down the bearing of broken lines on jump distance, which jointly correct nodes of *ADPH*.

(3)  Introducing a set of good points to improve the uniformity of the initial population of the *GWO*, introducing a flight strategy to ameliorate the convergence and exclude local optima of the *GWO* and using the improved *GWO* to evolve the coordinate positions of unknown nodes to improve the accuracy of place.

The rest of this paper is structured as follows. Section 2 briefs the related research of the localization algorithms. In Section 3, the localization procedure of *DV-Hop* and error analysis are detailed. *TWGDV-Hop* is presented in Section 4 and simulation results are discussed and localization performances are deliberated in Section 5. Finally, Section 6 presents conclusions and future research.

## 2. Related Research

Since its introduction, *DV-Hop* has been improved in many literatures. Generally, node localization consists of three steps: the first is improving the hops, the second is distance estimation and the third is coordinate estimation. When using these three steps to determine the coordinate of unknown nodes, error is inevitable, and the smaller the error, the higher the location accuracy. Recently, scholars improved the hops and distance estimation or used nature-inspired methods to optimize coordinate estimation to achieve a certain degree of accuracy improvement.

### 2.1. Improved Hops and Distance Estimation

Zhang et al. [28] proposed a beacon filtering-based localization algorithm that combines *DV-Hop* and multioutput support vector regression *MSVR*. The algorithm combines received signal strength indicator *(RSSI)*, *MSVR*, and weighted least squares *(WLS)* to estimate unknown node coordinates. Gao et al. [29] designed a modified localization algorithm that increases hops by introducing communication distance, calculates the optimal hop distance by using the weighted average of anchor nodes and the minimum mean square error criterion *(MMSE)*, and estimates node place by adopting *WLS*. Xue et al. [30] proffered a modified *DV-Hop*, which refines hops and emends distance. The minimum hops is emended by ushering *RSSI*, and *ADPH* is amended by the weighted *ADPH* error and estimated distance inaccuracy. Shikai [31] proffered a modified *DV-Hop* that uses the disparity between the genuine distance and evaluated the distance between beacon nodes in WSN to determine the corrected *ADPH* of beacon nodes and adopts two-dimensional hyperbolic functions to predict the place of unknown nodes. Hadir [32] proffered four new improved *DV-Hops*. *MSE* is employed to improve the jumping distance of anchor nodes, the two-dimensional hyperbola is employed to calculate the coordinates of unknown nodes, and particle swarm optimization *(PSO)* is employed to optimize the coordinates of unknown nodes. Jia et al. [33] proffered a novel localization algorithm in view of moving

anchor nodes and modifying hops. *ADPH* between the three closest anchor nodes to the unknown node is employed as the *ADPH* for the unknown node, and the position of different unknown nodes is calculated using multiple moving anchor nodes and their average value is the position of the unknown node. Lin et al. [34] proffered a distributed iterative refinement in view of correction vectors, which utilizes the fake stadiometry range and positioning range between nodes and their neighbors to construct a position-adjusted vector and a straightforward iterative hunt algorithm is used to figure out the minimum value of the sum of squares of the differences between these two ranges of the adjusted vector. Liouane et al. [35] proffered an improved *DV-Hop* algorithm that uses least squares localization methods and statistical filtering optimization strategies to reduce localization errors. Wan et al. [36] proffered a *WLS* loop *DV-Hop* combining the idea of optimal weight function and guideline picked anchors. Messous et al. [37] proffered a novel *DV-Hop* that used *RSSI* and polynomial approximation to gauge the space between unknown nodes and anchors and uses recursive calculations in the process of localization.

### 2.2. Use Nature-Inspired Methods to Optimize Coordinate Estimation

Huang et al. [38] proffered an advanced *DV-Hop*, which advances the weighted processing of anchor node's jump distance by introducing an *ADPH* error *Q-Tcd* and uses differential evolution (*DE*) to optimize the positioning results of unknown nodes. Liu et al. [39] proffered the *HDCDV-Hop* algorithm, which amended the evaluated distance between unknown nodes and various anchor nodes in view of fractional hops and anchor node coordinates and adopted an amended *DE* to obtain the evaluated position of unknown nodes. Shi et al. [40] provided an improved *DV-Hop*, which uses revised *PSO* and simulated an annealing hybrid algorithm to boost the positioning accuracy of the initial place of unknown nodes. Chen et al. [41] proffered *CWDV-Hop*, which calculates *ADPH* by ushering distance weighting factor and *ADPH* of single node. The position of unknown nodes is attained by using a two-dimensional hyperbolic format and optimized by using chicken swarm optimization algorithm (*CSO*). Cao et al. [42] proffered *DANSIDV-Hop*, which locates the dynamic anchor node set *(DANS)* based on binary particle swarm optimization (BPSO) and further optimizes the unknown node place using continuous *PSO*. Huang et al. [43] proffered *MA\*-3DV- Hop, which* optimizes the hops values of nodes and corrects the error of the *ADPH*. Yu et al. [44] proffered a novel *DV-Hop*, which uses a correction factor to correct hops, selects *ADPH* of unknown nodes based on the weight of each anchor node and calculates node coordinates using the cuckoo bird search algorithm. Sun et al. [45] proffered *GDV-Hop* that optimizes the place of unknown nodes by adopting modified *GWO* with adaptive strategies.

From the above analysis, all modified algorithms are based on the three processes of *DV-Hop*, but there are still certain defects and accuracy needs to be improved. This paper proposed a novel location algorithm based on *DV-Hop* and optimized *GWO* to improve the location accuracy of unknown nodes in *WSN*, to meet the needs of applications.

## 3. DV-Hop

### 3.1. The Process of DV-Hop

The process of *DV-Hop* is divided into three stages [24].

Step 1: Each anchor node spreads a group to the network with flood normal. All nodes acquire the minimum hops in this step.

Step 2: Calculate the *ADPH* for every anchor node by Equation (1).

$$HopS_i = \sum_{i \neq j} \sqrt{(x_i - x_j)^2 + (y_i - y_j)^2} / \sum_{i \neq j} H_{ij} \tag{1}$$

where $(x_i, y_i), (x_j, y_j)$ are the coordinates of nodes $i, j$. $H_{ij}$ is the minimum hops between the two nodes.

Then, spread $HopS_i$ to the network.

Step 3: Estimate the coordinate of unknown nodes.

The unknown node only receives *HopS* from the closest anchor node and calculates its distance between it and anchor nodes. For example, unknown node *N* first receives $HopS_i$, and the range between it and *i* is:

$$D_{ni} = HopS_i \times H_{ni} \tag{2}$$

Use the maximum likelihood method to list the equations, as shown in Equation (3):

$$\begin{cases} (x - x_1)^2 + (y - y_1)^2 = D_1{}^2 \\ (x - x_2)^2 + (y - y_2)^2 = D_2{}^2 \\ \qquad \vdots \\ (x - x_n)^2 + (y - y_n)^2 = D_n{}^2 \end{cases} \tag{3}$$

where $D_1, D_2, \cdots, D_n$ represent the estimated distance between *N* and each anchor node. The matrix expression of Equation (4) is: $AX = D$, where

$$A = \begin{bmatrix} (x_1 - x_n) & (y_1 - y_n) \\ (x_2 - x_n) & (y_2 - y_n) \\ \vdots & \vdots \\ (x_2 - x_n) & (y_2 - y_n) \end{bmatrix}, X = \begin{bmatrix} x \\ y \end{bmatrix}, D = \begin{bmatrix} x_1{}^2 - x_n{}^2 + y_1{}^2 - y_n{}^2 + D_n{}^2 - D_1{}^2 \\ x_2{}^2 - x_n{}^2 + y_2{}^2 - y_n{}^2 + D_n{}^2 - D_2{}^2 \\ \vdots \\ x_{n-1}{}^2 - x_n{}^2 + y_{n-1}{}^2 - y_n{}^2 + D_n{}^2 - D_{n-1}{}^2 \end{bmatrix}$$

$X = \left(A^T A\right)^{-1} A^T D$ can be obtained by using the least squares method.

### 3.2. Analyze Positioning Error of DV-Hop

The *DV-Hop* estimates the position of the unknown node by multiplying minimum hops between each node by *ADPH* of the anchor node through broadcast grouping of anchor nodes in the network. The error mainly comes from three aspects.

#### 3.2.1. Hops Error

When calculating the hops, the *DV-Hop* counts all neighboring nodes in the transmission range of the anchor node as one hop. As represented in Figure 3, let nodes B, C and D be one hop nodes in communication range of node A. Node E is outside communication range of A and can receive place information of anchor node forwarded by node D. Therefore, hops of node E are two. As shown in Figure 3, the different distance of nodes B and A, C and A, D, and A are relatively large, while the distance between E and A, and D and A is similar, but the hops are not the same. In this case, using traditional *DV-Hop* positioning algorithms will cause significant errors.

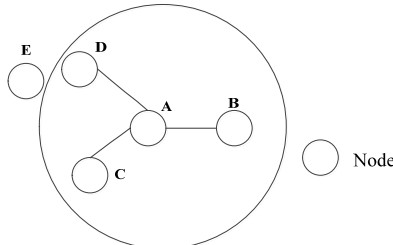

**Figure 3.** Hops Error Analysis.

#### 3.2.2. Error of *ADPH*

The localization accuracy of the *DV-Hop* mainly rests with whether *ADPH* is reasonable. In the case of multiple hops, let the hops between nodes *B* and *A* be four, and the line distance between *B* and *A* is approximated distance of the *DV-Hop* algorithm. It can be intuitively made out that there is a large deviation from the actual distance shown by the dashed line, as shown in Figure 4. Therefore, as hops increase, there is a significant localization error in areas with low node density.

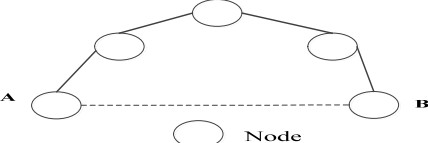

**Figure 4.** *ADPH* Error.

### 3.2.3. Error of Coordinate Estimation

Step three inevitably exhibits errors in step 2, which adds up when extracting the coordinate equation, begetting significant errors between localization results and the real place.

## 4. The TWGDV-Hop Algorithm

### 4.1. Correct Minimum Hops

When estimating hops between nodes by using the *DV-Hop*, all neighboring nodes in the communication region are considered as onehop. As shown in Figure 3 of Section 2.2 above, *B*, *C* and *D* are onehop nodes to *A*, yet their homologous true distances are distinct. If these nodes are dealt in the light of *DV-Hop*, it will result in significant positioning errors. To address such situations, first improve hops of nodes in the communication range of anchor nodes.

Broadcast messages by using anchor nodes with three different communication radii: $R/3$, $2R/3$, and $R$ and hops recorded as 1/3, 2/3 and 1. Divide the nodes in the region into a set of three regions, and the hops of unknown nodes are also real-coded accordingly. When the region is chopped up into three subareas in light of $R/3$, $2R/3$, and $R$, the unknown nodes are marked as virtual circles with $R/3$. As shown in Figure 5, the nodes $N_1$, $N_2$ are unknown nodes within the region and *A* is an anchor node. The cross zones between the virtual circle of unknown nodes and the anchor node are denoted as $S_{11}(1/3H)$, $S_{12}(1/3H)$, $S_{21}(2/3H)$, $S_{22}(2/3H)$, $S_{31}(H)$, and $S_{32}(H)$, where *H* is the hops.

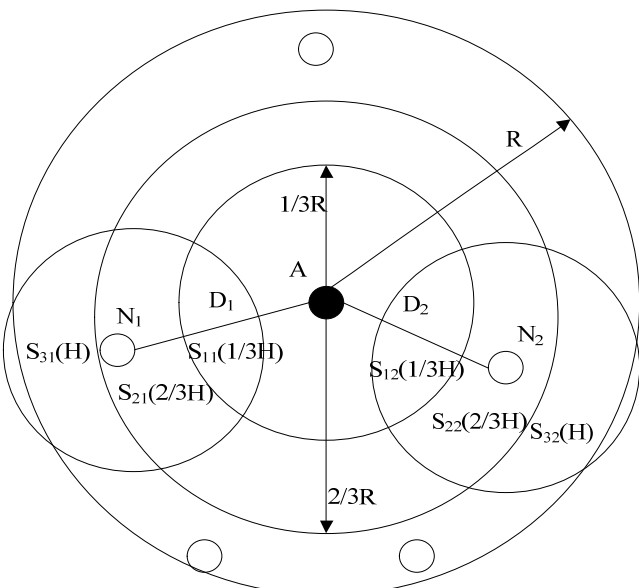

**Figure 5.** Regional Calculation Model.

Given the space between the node and anchor node, the zone of the intersecting space can be calculated. Assuming space between the node and the anchor node is

$D_i (i = 1, 2, \cdots, N)$, where $N$ is the number of anchor nodes, then adopting geometric Equations (4)–(6) to calculate the area of each intersecting domain [44].

$$
\begin{aligned}
S_{1i(1/3H)} &= R^2 \arccos \frac{(D_i{}^2 + R^2 + (1/9R)^2)}{2D_i R} \\
&+ (1/9R)^2 \arccos \frac{(D_i{}^2 + (1/9R)^2 - R^2)}{2D_i/9R} \\
&- \frac{1}{2}\sqrt{4D_i{}^2(1/9R)^2 - (D_i{}^2 - R^2 + (1/9R)^2)^2}
\end{aligned} \tag{4}
$$

$$
\begin{aligned}
S_{2i(2/3H)} &= R^2 \arccos \frac{(D_i{}^2 + R^2 + (1/3R)^2)}{2D_i R} \\
&- S_{1i(1/3h)} + (1/3R)^2 \arccos \frac{(3RD_i{}^2 + 1/3R - 3R^3)}{2D_i} \\
&- \frac{1}{2}\sqrt{4D_i{}^2(1/3R)^2 - (D_i{}^2 - R^2 + (1/3R)^2)^2}
\end{aligned} \tag{5}
$$

$$
\begin{aligned}
S_{3i(H)} &= \pi R^2 - R^2 \arccos \frac{(D_i{}^2 + R^2 + (1/3R)^2)}{2D_i R} \\
&+ (1/3R)^2 \arccos \frac{(3RD_i{}^2 + 1/3R - 3R^3)}{2D_i} \\
&- \frac{1}{2}\sqrt{4D_i{}^2(1/3R)^2 - (D_i{}^2 - R^2 + (1/3R)^4)}
\end{aligned} \tag{6}
$$

Assuming that nodes are unevenly arranged in an area with a total number of nodes $M$ and area $S$, the nodes in each subarea are denoted as $M_{i(1/3H)}$, $M_{i(2/3H)}$, and $M_{i(H)}$. Assume $\hat{S}_{1i(1/3H)}$, $\hat{S}_{2i(2/3H)}$, and $\hat{S}_{3i(H)}$ are the estimated areas of each region, calculated as shown in Equations (7)–(9).

The space between node and anchor node can be gained adopting the estimated area value. After that, a hop difference correction coefficient is introduced to correct the hops of nodes. Define the ratio of true distance $D_{ij}$ between nodes to the communication radius as the relative optimal hops.

$$
BH_{ij} = D_{ij}/R \tag{7}
$$

Contrast distinction between estimated hops $H_{ij}$ and $BH_{ij}$, and define the deviation coefficient $\gamma_{ij}$ using Equation (8).

$$
\gamma_{ij} = (H_{ij} - BH_{ij})/H_{ij} \tag{8}
$$

$\gamma_{ij}$ can mirror the distinction between $H_{ij}$ and $BH_{ij}$ between nodes that communicate with each other. The larger $\gamma_{ij}$ is, the greater the deviation between the two. Under the condition of a constant communication radius, $H_{ij}$ will be greater than or equal to $BH_{ij}$. For such situations, a difference correction coefficient $\eta_{ij}$ is defined by using Equation (9) to optimize hops and reduce the accumulation of errors.

$$
\eta_{ij} = 1 - \gamma_{ij}{}^n \tag{9}
$$

By using Equation (10), the modified hops between nodes can be obtained.

$$
H'_{ij} = \eta_{ij} H_{ij} \tag{10}
$$

The corrected hops are closer to the relative optimal hops, and the error in hops will be smaller.

### 4.2. Correct the ADPH of Nodes

The original *DV-Hop* algorithm adopted unbiased estimation [46] to compute *ADPH*. The fitness function is as follows:

$$
fit = \frac{1}{N-1}\sum_{i \neq j}(D_{ij} - HopS_i \times H_{ij}) \tag{11}
$$

If $fit = 0$, is computed as:

$$HopS_i = \sum_{i \neq j} D_{ij} / \sum_{i \neq j} H_{ij} \tag{12}$$

where $H_{ij}$ is the hops between nodes $i$ and $j$, and both have $i \neq j$, $D_{ij}$ is distance between $i$ and $j$. However, this approach cannot accommodate that the error is randomly distributed and positive and negative phases, which leads to the minimization of global error but not the minimization of all errors. Therefore, the square error criterion is adopted in this paper to minimize the overall error. The fitness function is:

$$fit = \frac{1}{N-1} \sum_{i \neq j} (D_{ij} - HopS_i \times H_{ij})^2 \tag{13}$$

Take the partial derivative of $HopS$ for $fit$ and take it as zero to obtain $HopS_i'$.

$$HopS_i' = \frac{\sum_{i \neq j} D_{ij} H_{ij}}{\sum_{i \neq j} H_{ij}^2} \tag{14}$$

However, when computing $ADPH$ of anchor nodes, it does not think about the distance of the broken line is the notional distance, as represented in Figure 4, which clearly has a significant error compared to the true distance shown by the dashed line. Therefore, this article refers to reference [36] and introduces a distance weighting factor to further modify to reduce impact of the line distance on $ADPH$.

According to Equation (1), we can infer:

$$
\begin{aligned}
HopS_i &= \frac{\sum_{i \neq j} \sqrt{(x_i - x_j)^2 + (y_i - y_j)^2}}{\sum_{i \neq j} H_{ij}} \\
&= \frac{\sum_{i \neq j} D_{ij}}{\sum_{i \neq j} H_{ij}} = \frac{D_{i1} + D_{i2} + \cdots + D_{iM}}{H_{i1} + H_{i2} + \cdots + H_{iM}} \\
&= \frac{H_{i1}}{\sum_{i \neq j} H_{ij}} \times \frac{D_{i1}}{H_{i1}} + \frac{H_{i2}}{\sum_{i \neq j} H_{ij}} \times \frac{D_{i2}}{H_{i2}} + \cdots + \frac{H_{iM}}{\sum_{i \neq j} H_{ij}} \times \frac{D_{iM}}{H_{iM}} \\
&= \phi_{i1} \times \frac{D_{i1}}{H_{i1}} + \phi_{i2} \times \frac{D_{i2}}{H_{i2}} + \cdots \phi_{iM} \times \frac{D_{iM}}{H_{i2}} \\
&= \sum_{j=1}^{M} \left( \frac{H_{ij}}{\sum_{i \neq j} H_{ij}} \times \frac{D_{ij}}{H_{ij}} \right) = \sum_{j=1}^{M} (\phi_{ij} \times HopS_{ij})
\end{aligned} \tag{15}
$$

where $M$ is the number of beacon nodes. $\varphi_{ij}$ is denoted as the hop-weighted factor.

As shown in Figure 6, Hops between anchor node A and B are $H_{AB}$, $\varphi_{ij}$ increases as it increases. The distance between the two nodes has curved, changing from a direct line to a polyline, resulting in a significant error from the true distance. To reduce the impact of this error on $ADPH$, a distance weighting factor is introduced. Transform Equation (14) into Equation (15):

$$
\begin{aligned}
HopS_i'' &= \frac{\sum_{i \neq j} \sqrt{(x_i - x_j)^2 + (y_i - y_j)^2}}{\sum_{i \neq j} H_{ij}} \\
&= \frac{\sum_{i \neq j} D_{ij}}{\sum_{i \neq j} H_{ij}} = \frac{D_{i1} + D_{i2} + \cdots + D_{iM}}{H_{i1} + H_{i2} + \cdots + H_{iM}} \\
&= \frac{D_{i1}}{\sum_{i \neq j} D_{ij}} \times \frac{D_{i1}}{H_{i1}} + \frac{D_{i2}}{\sum_{i \neq j} D_{ij}} \times \frac{D_{i2}}{H_{i2}} + \cdots + \frac{D_{iM}}{\sum_{i \neq j} D_{ij}} \times \frac{D_{iM}}{H_{iM}} \\
&= \delta_{i1} \times \frac{D_{i1}}{H_{i1}} + \delta_{i2} \times \frac{D_{i2}}{H_{i2}} + \cdots \delta_{iM} \times \frac{D_{iM}}{H_{i2}} \\
&= \sum_{j=1}^{M} \left( \frac{D_{ij}}{\sum_{i \neq j} D_{ij}} \times \frac{D_{ij}}{H_{ij}} \right) = \sum_{j=1}^{M} (\delta_{ij} \times HopS_{ij})
\end{aligned} \tag{16}
$$

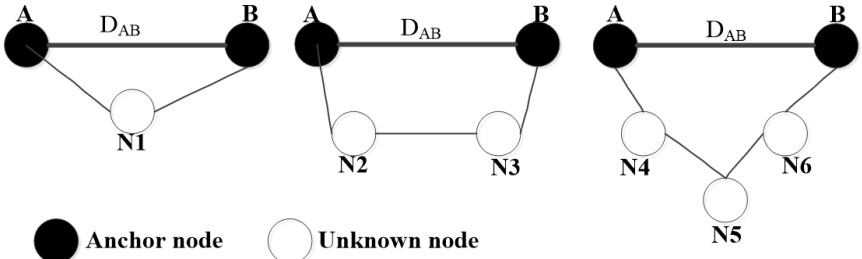

**Figure 6.** Graph of increases.

$\delta_{ij}$ is denoted as the distance weighting factor (*DWF*). From the figure, it can be seen that $\delta_{ij}$ is not affected by the increase in $H_{AB}$, which implies the *DWF* can diminish influence of twisted paths on *ADPH*.

To avoid that a single *ADPH* cannot reflect the *ADPH* of the entire network situation well and to diminish error accumulation brought by inaccurate data, *ADPH* computed by adopting the square error criterion and corrected by introducing a distance weighting factor will be equalized twice to obtain the corrected one.

$$AVGHopS_i = \frac{HopS_i' + HopS_i''}{2} \tag{17}$$

Using Equation (17) to obtain the corrected *ADPH* can be closer to tangible *ADPH* in the whole network, further reducing the influence of hop distance error accumulation and improving positioning accuracy.

### 4.3. Optimization of GWO
#### 4.3.1. Standard GWO

The *GWO* is a biomimetic population algorithm for optimization proffered by Australian researcher Mirjalili et al. [47]. It has preponderances of strong convergence, minor parameters and being likely to be implemented, which makes it a research hot spot in localization algorithms in recent years.

Grey wolves belong to the canine family of social animals. They strictly adhere to social hierarchical dominance at the peak of the food chain, as shown in Figure 7.

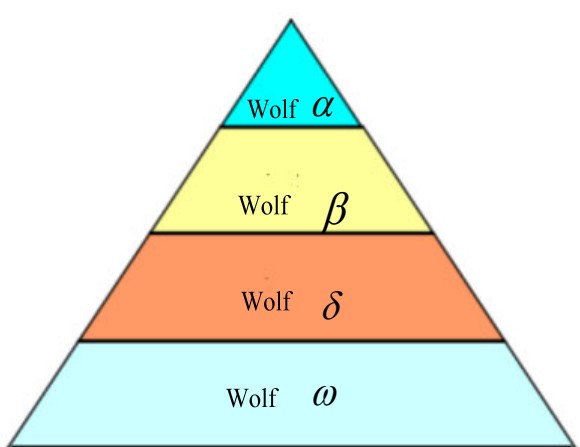

**Figure 7.** Gray wolf hierarchy diagram.

The grey wolf with the highest social level is called $\alpha$, which has the highest ruling status and absolute dominance over other wolves. The second- and third-ranked gray wolves in a social hierarchy are called $\beta$ wolf and $\delta$ wolf. The lowest-level wolf in society is called $\omega$ wolf, which needs to obey the other levels of wolves with the role of preventing self-killing within wolf packs. The stages of GWO are [47]:

(1) Social Hierarchy. Tag the three wolves with the best fitness as $\alpha$ wolf, $\beta$ wolf, and $\delta$ wolf when designing the grey wolf algorithm. The remaining gray wolves are called $\omega$ wolf. GWO optimization is completed by the process of iteratively updating the best three solutions $\alpha$, $\beta$ and $\delta$.

(2) Encircling prey. When gray wolf packs search for prey, they gradually form a circle to surround the quarry target. The mathematical pattern of the deed is:

$$\begin{aligned} D &= |C * Xp(t) - X(t)| \\ X(t+1) &= Xp(t) - A * D \end{aligned} \tag{18}$$

where $t$ and $t+1$ express two consecutive iterations, * is the Hadamard product operator, $X_p$ is the location of optimization objective; and $X_t$ represents the positioning vector of the wolf at this time. $A$ and $C$ are synergistic coefficient vector values; and D represents the distance between gray wolves. The diminished range between the wolves and their quarry count upon vectors $A$ and $C$. They are computed as below:

$$\begin{aligned} A &= 2a * r_1 - a \\ C &= 2r_2 \end{aligned} \tag{19}$$

From the beginning to the end of the iteration, A lessens linearly from two to zero; B and $r_2$ are stochastic vectors in 0~1.

(3) Venery. Grey wolves have the talent to distinguish potential target localization and the search process is mainly led by $\alpha$, $\beta$ and $\delta$ wolf. However, due to the unknown characteristics of the solution space, the wolf pack cannot determine the perfect positioning of the optimal target. To better simulate the chasing deed of gray wolves, it is presumed that $\alpha$, $\beta$ and $\delta$ have good discriminative target localization talents, which are retained during each iteration. Then, update the place of the $\omega$ wolf. The mathematical model for this act is as below:

$$\begin{cases} D_\alpha &= |C_1 * X_\alpha(t) - X(t)| \\ D_\beta &= |C_2 * X_\beta(t) - X(t)| \\ D_\delta &= |C_3 * X_\delta(t) - X(t)| \end{cases} \tag{20}$$

$$\begin{cases} X_1 = X_\alpha - A_1 * D_\alpha \\ X_2 = X_\beta - A_2 * D_\beta \\ X_3 = X_\delta - A_3 * D_\delta \end{cases} \tag{21}$$

$$X(t+1) = \frac{1}{3}(X_1 + X_2 + X_3) \tag{22}$$

where $X_\alpha$, $X_\beta$ and $X_\delta$ are the azimuth vectors of $\alpha$, $\beta$ and $\delta$ in the gray wolf pack. X stands for the place of the gray wolf. $D_\alpha$, $D_\beta$ and $D_\delta$ and C, respectively, represent the distances between $\alpha$, $\beta$, and $\delta$ and the current gray wolf. When $|A| > 1$, members of the gray wolf pack disperse in various areas and search for targets. When $|A| < 1$, members of the gray wolf pack will concentrate on searching for prey targets in a specific area.

From Figure 8, we can make out that the final place of the candidate wolf is located within the stochastic circle positions defined by $\alpha$, $\beta$ and $\delta$. $\omega$ wolf is behind the general orientation of the prey target predicted by the three higher-order gray wolves $\alpha$, $\beta$ and $\delta$ at this time, which is the position of any new higher-order gray wolf around the prey.

(4) Attacking target. When creating the model, according to step 2, $a$ decrease will change $A$, where $A$ is a random vector in $[-a, a]$. When $A$ is on interval $[-a, a]$, the place between the gray wolf and the target is the next moment of the agent search.

(5) Look for prey. Mainly relying on $\alpha$, $\beta$ and $\delta$ three high-order gray wolves to search for prey. They disperse to find the target direction of the prey and then focus on attacking.

In the decentralized model, when $|A| > 1$, the agent search is kept away from the prey, which allows the algorithm to perform global optimization.

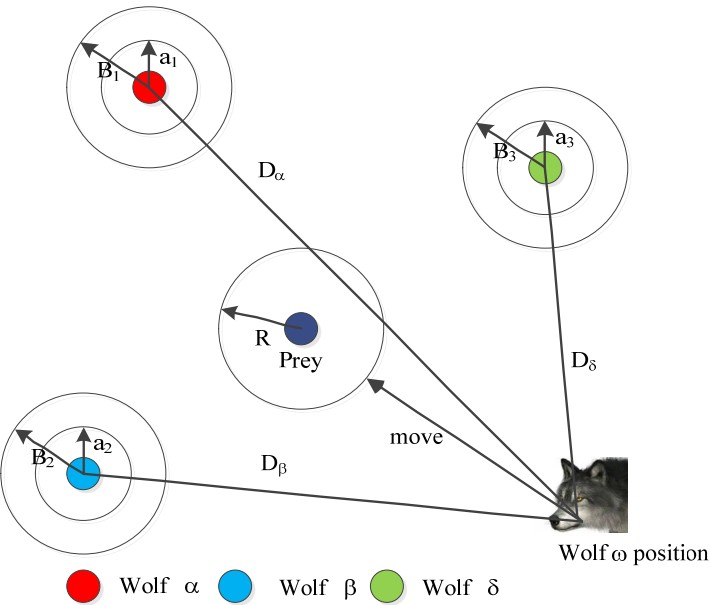

**Figure 8.** Diagram of choose wolf movement hen hunting.

*GWO* is a new bionic collective smarts algorithm that simulates the gray wolf group hunting heuristic with the advantage of a high convergence performance and being easy to implement, which means it can be used for *WSN* node location to reduce algorithm error. But like many bionic algorithms, it has problems, for instance, immature convergence and frail global search faculty, which lead to inferior accuracy of the final solution.

### 4.3.2. The Improved GWO

(1)  Initialize the population of the best point set

The quality of the initial population distribution immediately affects the convergence and level of *GWO*. An initial population with good distribution characteristics can make the majorization performance of it better. In the *GWO* algorithm, the initialization of the population adopts a random distribution, which cannot fully cover the space of the solution, so it is difficult to traverse various possible situations of the solution, which can affect the quality of the final solution due to premature convergence. However, this article uses the set of good points proposed by Hua et al. for population initialization to avoid this problem [48].

Presuming a population size of 100, Figures 9 and 10 are populations initialized using stochastic distribution and a set of good points, respectively. We can find Figure 10 is evener, which sufficiently swathes the understanding space and increases the diversity of the population, so the algorithm will have better ergodicity. In theory, it has been proven that the weighted sum of *n* good points yield smaller errors than any other *n* points.

(2)  Levy Flight Strategy

Because *GWO* has the characteristics of premature convergence and low global optimization ability, the *Levy* flight algorithm with the random walk performance is integrated into *GWO* to upgrade the global optimization ability of the *GWO* and enrich the population discrepancy of *GWO* in this paper.

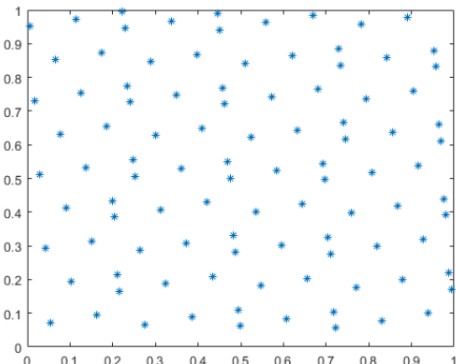

**Figure 9.** Initial Population of the Best Point Set.

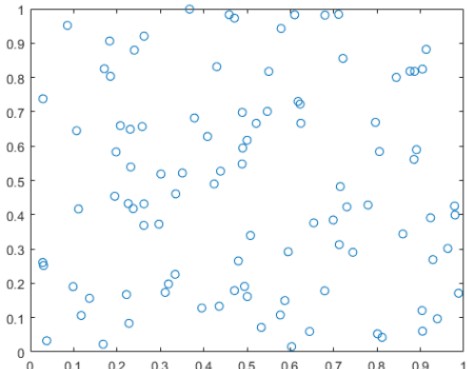

**Figure 10.** Initial Population of Random Distribution.

*Levy* flight is a special stochastically wander strategy [49] that adopts the random walk method and focuses on short distance search and occasionally conducts long distance search, which is shown in Figure 11. Using this strategy in swarm intelligence algorithms can make individuals widely distributed in the search space when searching for the optimal solution in a large range, increase multiplicity of the population, boost the whole optimization capacity and avert sinking into the localized optimal solution prematurely.

*Levy* flight strategy follows the *Levy* distribution, usually represented by a power-law distribution: $L(S) \sim |S|^{-1-\beta} (0 < \beta < 2)$, where $S$ is step, and $L(S)$ is the probability of moving step $S$. Due to the complexity of *Levy* distribution, the *Mantegna* algorithm is usually used to simulate it [49]:

$$S = \frac{\theta}{|\vartheta|^{1/\beta}} \tag{23}$$

where $\theta$ and $\vartheta$ follow a normal distribution [49]:

$$\theta \sim N\left(0, \sigma^2\right), \; \vartheta \sim N(0, 1) \tag{24}$$

$$\sigma = \left\{ \frac{\tau(1+\beta)\sin(\pi\beta/2)}{2^{(\beta-1)/2}\tau[(1+\beta)/2]\beta} \right\}^{1/\beta} \tag{25}$$

where $\tau$ is a standard gamma function and $\beta$ usually takes a value of 1.5.

Although this strategy boosts the global optimization ability of the algorithm, if all individuals use this strategy to update positions in each iteration, the amount of calculation will be greatly increased. Therefore, when the fitness value does not change significantly for consecutive *limit* times (that is, the change value is less than 0.0001), we determine

that the algorithm is trapped in a local optimum, and then use *Levy* flight Equation (26) to perform a flight optimization.

$$X_i^{t+1} = X_i^t + \varphi L(S) = X_i^t + \varphi \times \frac{\theta}{|\vartheta|^{1/\beta}} \tag{26}$$

where $\phi$ is a step scaling factor, as shown in the Equation (27).

$$\varphi = 0.02(X_i - X_j) \tag{27}$$

where $X_i$ and $X_j$ are arbitrarily different solutions.

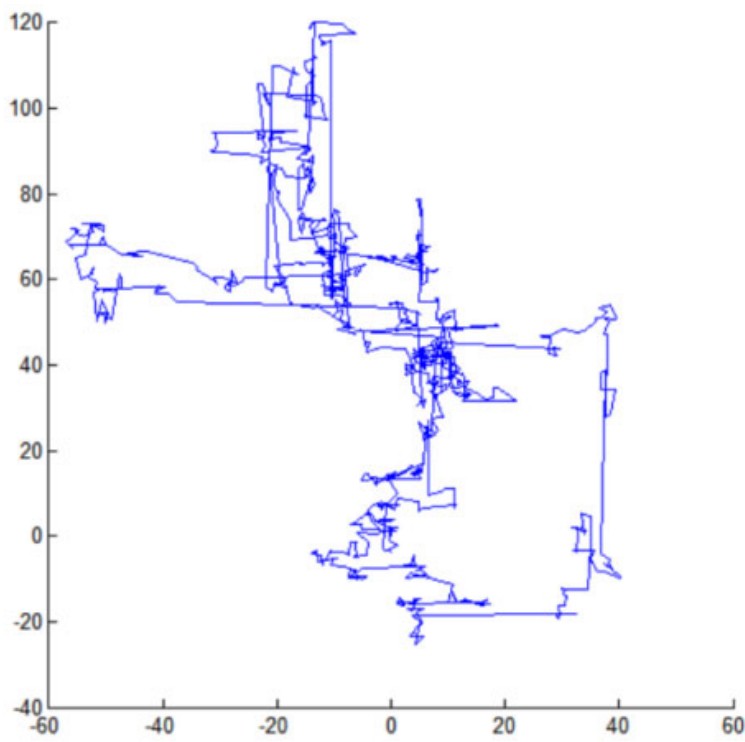

**Figure 11.** Levy Flight Strategy Simulation.

### 4.3.3. The Improved *GWO* Correct Unknown Node Positions

The third stage of this algorithm is identical to *DV-Hop*. Use the maximum likelihood estimation algorithm to determine the coordinate position of the desired node. Because of the limitations of this method, it cannot effectively improve the calculation accuracy. Therefore, *TWGDV-Hop* adds a fourth stage, which uses the improved *GWO* to optimize the position of the unknown nodes obtained. The steps are:

Step 1: Initialize three high-order gray wolves α, β and δ. Set the relevant parameters including population size $N$, maximum number of iterations *Tmax* and continuous number of iterations *limit*. Initialize the grey wolf population using the set of good points from the previous section $\{X_i : X_1, X_2, \cdots, X_N\}$ and make the current $t = 0$.

Step 2: Calculate the fitness values of all gray wolves according to Equation (28) and sort them. Set the top three ranking results as α, β and δ, respectively.

$$fit = \frac{1}{N}\sum_{i=1}^{N}\left| \sqrt{(x - x_i)^2 + (y - y_i)^2} - D_i \right| \tag{28}$$

where $(x, y)$ *and* $(x_i, y_i)$ are the coordinates of the unknown node and the anchor node, respectively. $D_i$ is the actual distance between the unknown node and the anchor node.

Step 3: Use Equations (19)–(22) to update the space of the gray wolf population and update the coefficient vector *A*, *C*.

Step 4: Judge whether the algorithm is trapped in the local optimum. If it iterates *limit* times in succession, the fitness value does not change significantly, and *Levy* flight strategy is ushered in the gray wolf population.

Step 5: Determine whether the algorithm has clocked up the stop condition. If the stop condition is clocked up, the output position of *α* wolf is the final improved place of the unknown node.

## 5. Simulation Experiments and Result Analysis

### 5.1. Simulation Experiment Settings

For verifying the performance of *TWGDV-Hop* proffered in this paper, simulations were conducted in Matlab2014a to reckon and study their positioning errors and accuracy. Through simulation, the validity of our proffered algorithm was compared with *DV-Hop*, *WDV-Hop*, *CWDV-Hop* proposed in reference [41], *HWDV-HopPSO* proposed in reference [32] and *GDV-Hop* proposed in reference [45].

To diminish the influence of stochastic errors on the results in the tentative experiment, the final result is the mean of 100 tentative experiments to evaluate the localization accuracy and localization error of every way. Three distinct network topologies were chosen, as shown in Figure 12. Namely, square random, O-shaped, and X-shaped. Square random is one of the most used topologies in *WSN*, in which nodes are stochastically distributed in square regions. The O-shaped topology is an irregular network topology, while the X-shaped topology is also based on the random deployment of nodes in the four sides empty area. For each topology, the system parameters are set as shown in Table 1. The network deployment area is a two-dimensional plane of 500 m × 500 m. All nodes with the same structure and communication radius are randomly arranged to form a self-forming network.

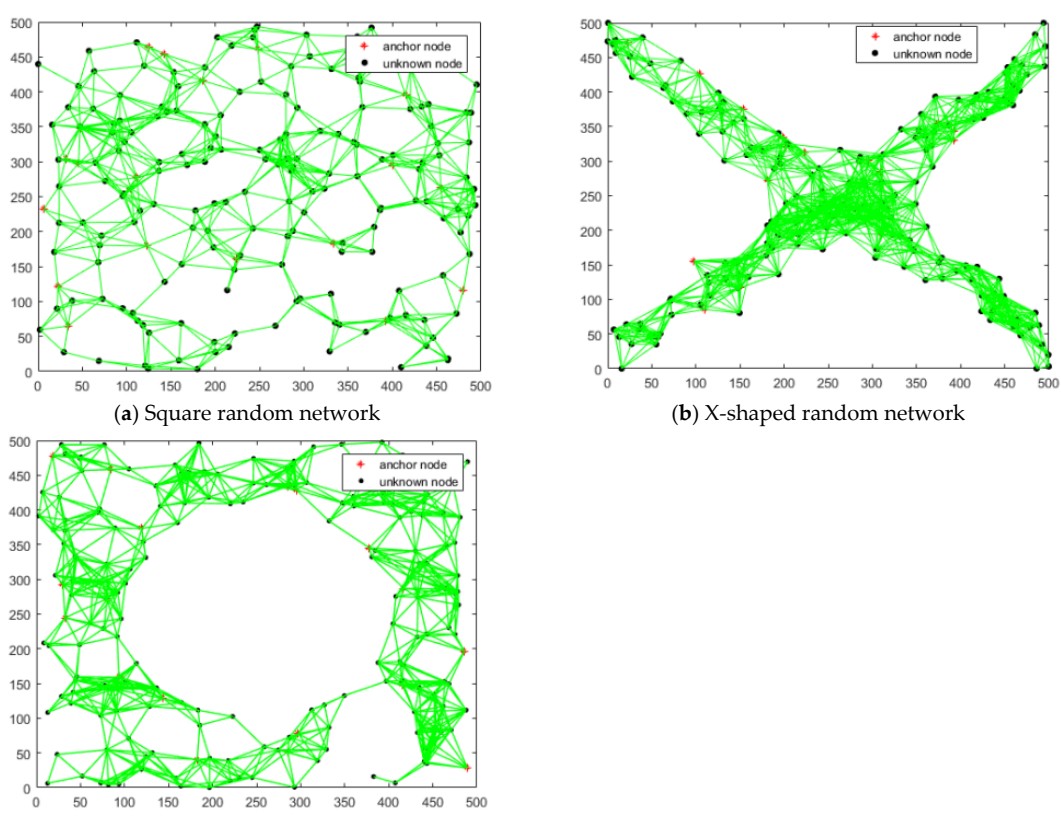

(**a**) Square random network

(**b**) X-shaped random network

(**c**) O-shaped random network

**Figure 12.** Nodes deployment according to network topologies.

**Table 1.** Simulation Parameter Settings.

| Parameter | Value |
|---|---|
| Network | |
| Network topology | Square random, O-shaped, X-shaped |
| Total runs | 100 |
| Length of area | $500 \times 500$ |
| Total number of nodes | 300, 350, 400, 450, 500 |
| Number of anchor nodes | 30, 35, 40, 45, 50, 55, 60 |
| Communication range R | 60, 70, 80, 90, 100 |
| GWO | |
| Number of iterations | 100 |
| Size of wolfs | 80 |
| Limit | 70 |
| a | $2-0$ |
| r1, r2 | Rand [0, 1] |

We change parameters based on three network topologies, such as the number of anchor nodes, the total number of sensor nodes and the communication radius of nodes, to explore the performance of algorithms under distinct network topologies. The Localization Accuracy (*LA*) is adopted to evaluate the function of the algorithm. It can be expressed [32] as follows:

$$LA = \frac{\sum_{i=1}^{N} \sqrt{(x_t^i - x_e^i)^2 + (y_t^i - y_e^i)^2}}{N \times R} \times 100\% \qquad (29)$$

where $(x_t^i, y_t^i)$ is the actual coordinates of the unknown node. $(x_e^i, y_e^i)$ are the estimated coordinates. $N$ is the number of unknown nodes.

The localization error (*LE*) reflects the deviation between the true position of unknown nodes and the estimated coordinates. The equation is [30]:

$$LE = \sqrt{(x_t^i - x_e^i)^2 + (y_t^i - y_e^i)^2} \qquad (30)$$

*5.2. Analysis of Simulation Results*

(1) The impact of the number of anchor nodes

The total number of nodes is fixed at 300. The communication radius is 60 m and the number of anchor nodes have grown from 30 to 60. The simulation results are manifested in Figures 13a, 14a and 15a.

Typically, the trend of LA of most localization algorithms diminishes little by little along with the number of anchors added in the three networks. The *LA* of *DV-Hop, WDV-Hop, CWDV-Hop, HWDV-HopPSO, GDV-Hop,* and *TWGDV-Hop* is approximately 0.52R, 0.42R, 0.27R, 0.32R, 0.34R, and 0.25R, respectively. Compared with other algorithms, the proposed *TWGDV-Hop* achieves a lower localization error. The localization accuracy of TWGDV-Hop algorithm has been improved by approximately 51.6%, 39%, 7%, 23%, and 24%, respectively. The localization accuracy of *DV-Hop, WDV-Hop, CWDV-Hop, HWDV-HopPSO, GDV-Hop* and *TWGDV-Hop* is approximately 1.01R, 0.62R, 0.61R, 0.72R, 0.62R and 0.59R, respectively, in the X-shaped network, and the *LA* of *TWGDV-Hop* has been improved by approximately 42.2%, 9.4%, 4.7%, 18.9% and 5.8%, respectively. The localization accuracy of *DV-Hop, WDV-Hop, CWDV-Hop, HWDV-Hop, GV-Hop* and *TWGDV-Hop* is approximately 1.15R, 1.11R, 0.39R, 0.62R, 0.60R and 0.36R, respectively, in the O-shaped network, and the localization accuracy of *TWGDV-Hop* has been improved by approximately 68.6%, 67.6%, 6.9%, 41.9% and 40.2%, respectively.

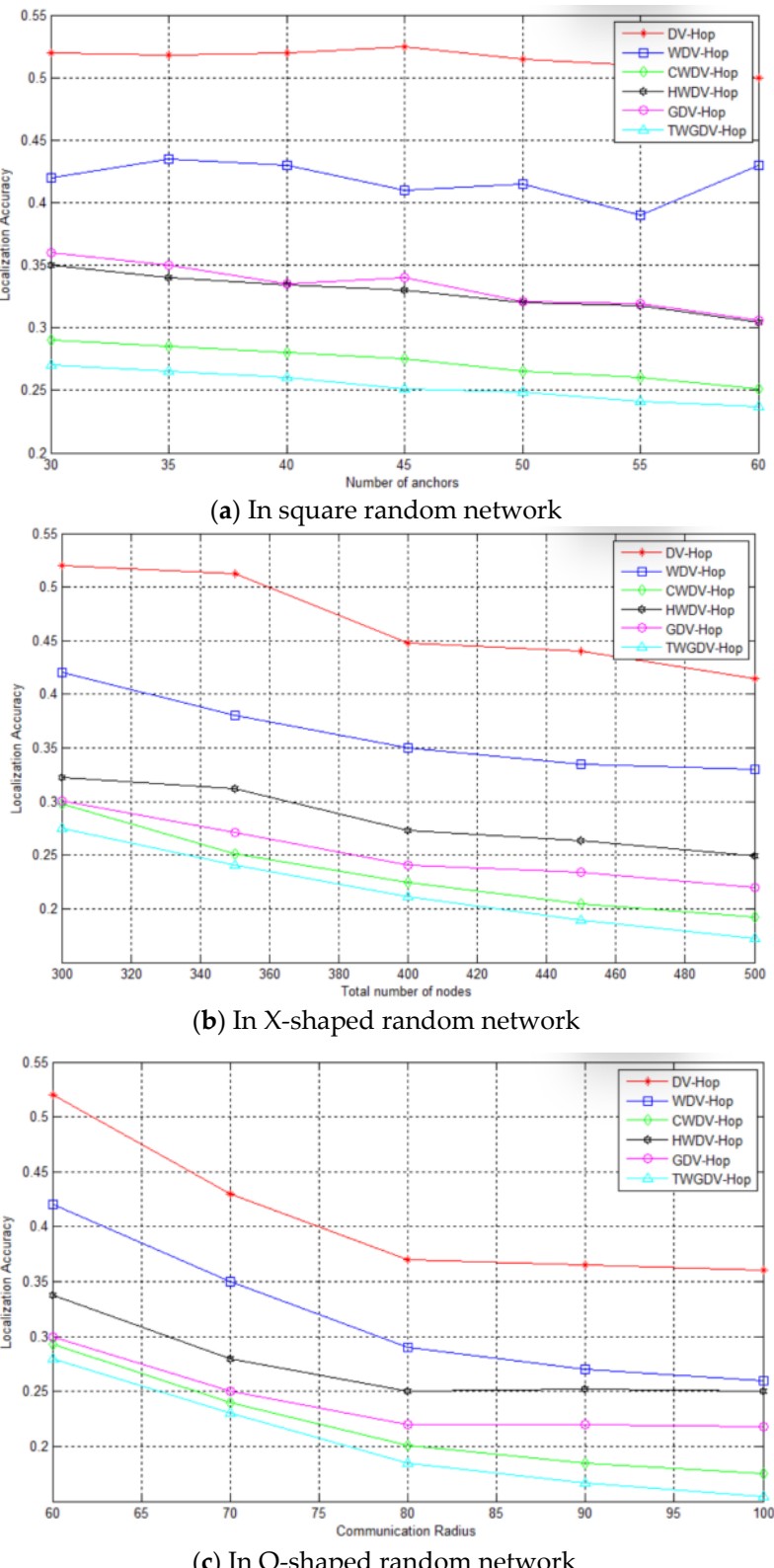

(**a**) In square random network

(**b**) In X-shaped random network

(**c**) In O-shaped random network

**Figure 13.** The impact of the number of anchor nodes.

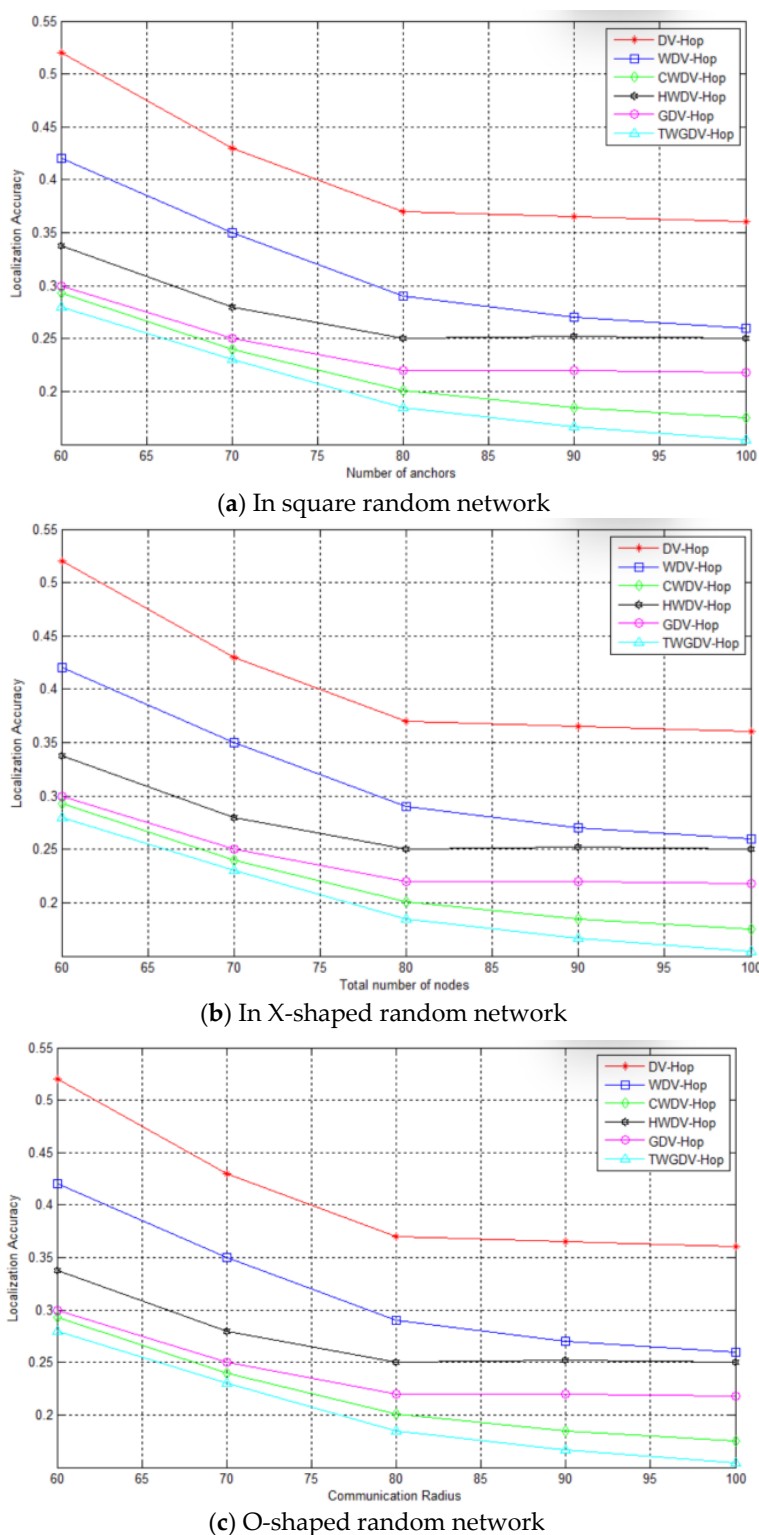

(**a**) In square random network

(**b**) In X-shaped random network

(**c**) O-shaped random network

**Figure 14.** The impact of the total number of nodes.

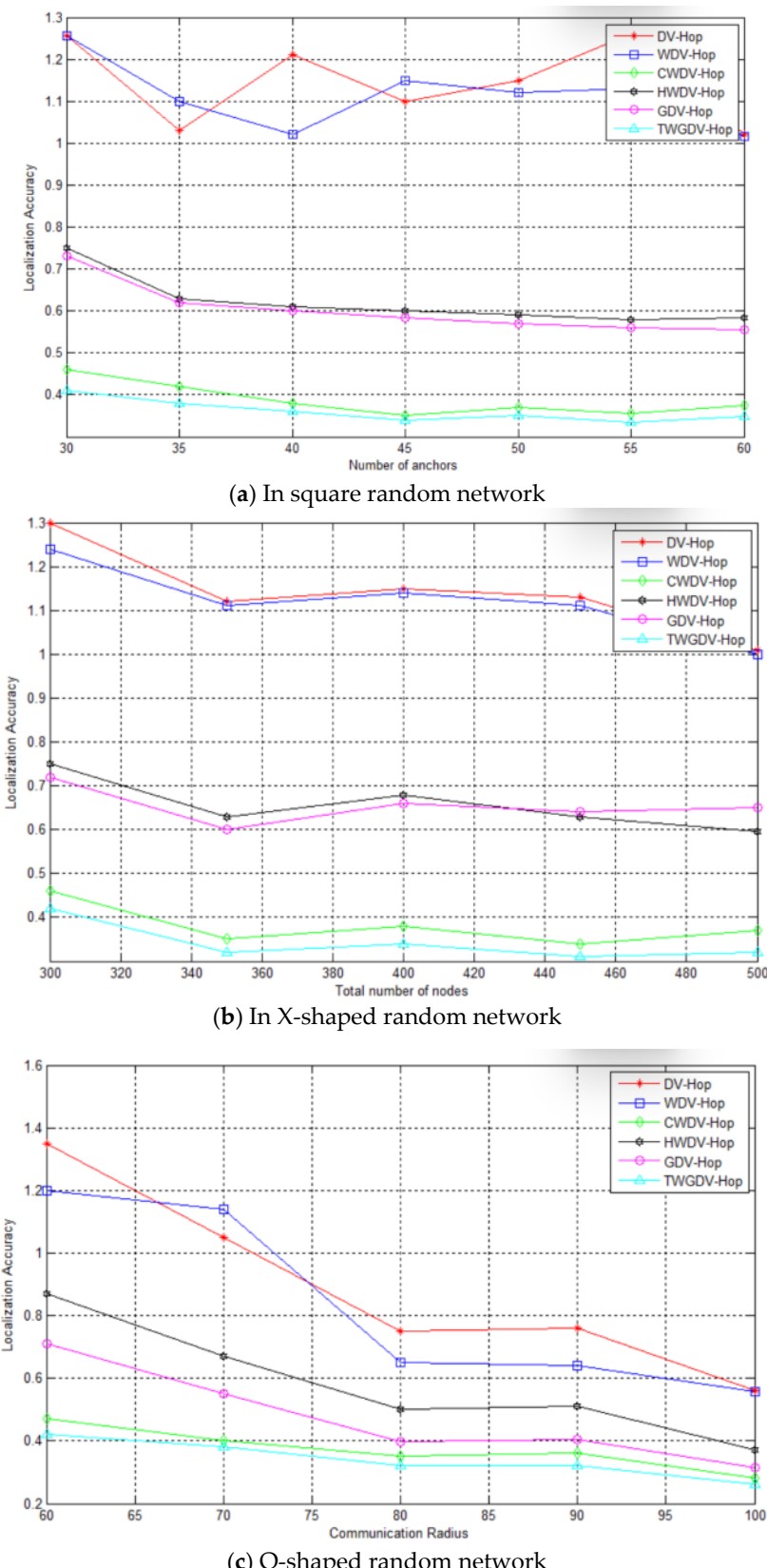

(**a**) In square random network

(**b**) In X-shaped random network

(**c**) O-shaped random network

**Figure 15.** The impact of the communication radius.

(2)    The Influence of the total number of nodes

The fixed communication radius is 60 m, the proportion of anchor nodes is 10%, and the total number of nodes has been increased from 300 to 500, which are shown in Figures 13b, 14b and 15b. When the density altitude of the network increases, the performance is greatly improved because the network has higher connectivity as many master nodes are deployed. The *LA* of *DV-Hop, WDV-Hop, CWDV-Hop, HWDV-HopPSO, GDV-Hop* and *TWGDV-Hop* is approximately 0.47R, 0.36R, 0.23R, 0.28R, 0.25R, and 0.21R, respectively, in a square network. The *LA* of *TWGDV-Hop* has been raised by 53.4%, 40.1%, 33.9%, 7.1%, 23.4%, and 14.2%, respectively. The *LA* of *DV-Hop, WDV-Hop, CWDV-Hop, HWDV-HopPSO, GDV-Hop* and *TWGDV -Hop* is approximately 1.01R, 0.66R, 0.63R, 0.72R, 0.64R, and 0.61R, respectively, in the X-shaped network. The *LA* of the *TWGDV-Hop* has been boosted by 39.7%, 8.4%, 3.1%, 15.2%, and 4.7%., respectively. It increased by 70.1%, 69.5%, 10%, 48%, and 47.7%, respectively, in the O-shaped network. Obviously, *TWGDV-Hop* surpasses other algorithms and exhibits better positioning accuracy.

(3)    The impact of the communication radius

The whole number of nodes is fixed at 300. The proportion of anchor nodes is 10% and the communication radius has been increased from 60 m to 100 m. The results are presented in Figures 13c, 14c and 15c. The results show that as the network becomes more connected, the localization errors of the three networks decrease with communication range growing. In the square network, compared with the other algorithms, the *LA* of *TWGDV-Hop* has been improved by 50.3%, 36.1%, 7.2%, 25.8% and 15.9%, respectively. In the X-shaped random network, the *LA* has been improved by 38.9%, 4.3%, 2.4%, 9.3% and 5.1%, respectively. In the O-shaped network, the *LA* has been improved by 61.9%, 59.4%, 8.5%, 41.7%, and 28.4%, respectively.

We can make out that *TWGDV-Hop* method has better *LA* than other methods in isotropic and anisotropic networks from the simulation results because the improved algorithm uses a hop difference correction coefficient to make the minimum hos between nodes more accurate. Introducing a distance-weighting factor to modify the *ADPH* of anchor nodes effectively alleviates the impact of curve paths on the *ADPH* and adopting an improved *GWO* makes the coordinate positions of unknown nodes more accurate.

(4)    Localization error analysis

The localization errors of all above algorithms are presented in Tables 2–4. Nodes are stochastically laid out in an area of 500 × 500. There are 30 anchor nodes and a communication radius of 60 m for nodes. We have compared our improved algorithm with other algorithms in minimum, maximum, and average localization errors.

**Table 2.** The minimum, maximum and average localization errors in square random network.

| Algorithm | Min | Max | Mean |
|---|---|---|---|
| DV-Hop | 1.78 | 92.31 | 35.78 |
| WDV-Hop | 0.49 | 88.08 | 23.17 |
| CWDV-Hop | 0.51 | 88.94 | 16.23 |
| HWDV-HopPSO | 0.8 | 77.73 | 24.48 |
| GDV-Hop | 0.83 | 84.01 | 23.18 |
| TWGDV-Hop | 0.49 | 82.21 | 15.79 |

It can be made out that the LE of *TWGDV-Hop* is smaller than that of other localization algorithms. This is because the hop deviation factor is introduced in the *TWGDV-Hop* algorithm to cause hops between nodes to be more accurate and the distance-weighting method introduced greatly reduces the impact of curved paths on *ADPH*. In addition, the improved *GWO* applied to *TWGDV-Hop* also effectively optimized the coordinate accuracy of unknown nodes. Based on these three advantages, the proffered *TWGDV-Hop* has better localization performance than the other five algorithms.

**Table 3.** The minimum, maximum and average localization errors in X-shaped random network.

| Algorithm | Min | Max | Mean |
|---|---|---|---|
| DV-Hop | 1.37 | 163.52 | 63.01 |
| WDV-Hop | 3.58 | 131.76 | 39.57 |
| CWDV-Hop | 0.82 | 71.62 | 39.01 |
| HWDV-HopPSO | 0.52 | 127.3 | 50.01 |
| GDV-Hop | 0.84 | 76.55 | 43.25 |
| TWGDV-Hop | 0.76 | 68.13 | 36.89 |

**Table 4.** The minimum, maximum and average localization errors in O-shaped random network.

| Algorithm | Min | Max | Mean |
|---|---|---|---|
| DV-Hop | 7.42 | 139.26 | 61.02 |
| WDV-Hop | 2.01 | 130.31 | 53.65 |
| CWDV-Hop | 1.14 | 81.8 | 23.56 |
| HWDV-HopPSO | 1.4 | 85.63 | 36.98 |
| GDV-Hop | 1.54 | 103.58 | 29.92 |
| TWGDV-Hop | 1.09 | 77.8 | 22.54 |

(5)    Time complexity analysis

In WSN, sensor nodes have limited computing power and energy, so the time complexity of the algorithm should be considered. Hypothesizing that the total number of sensor nodes in WSN is $S$, the number of anchor nodes is $M$, the maximum iterations for *CSO, PSO* and *GWO* are $I_1$, $I_2$, and $I_3$, respectively, and the population size is $P_1$, $P_2$ *and* $P_3$, respectively. *DV-Hop*, *WDV-Hop*, *CWDV-Hop*, *HWDV-Hop*, *GDV-Hop* and *TWGDDV-Hop* algorithms all need to calculate the minimum hops matrix, so the time complexity is $O(S^2)$. In step 2, *DV-Hop* and *GDV-Hop* use the same way to calculate *ADPH* of anchor nodes, resulting in a time complexity of $O(M^2)$. Although other proposed algorithms, such as *WDV-Hop*, *HWDV-HopPSO*, *CWDV-Hop* and *TWGDV-Hop* use different weighted techniques, their time complexity is $O(M^2)$. In step three, although the six algorithms use different methods to estimate the coordinate positions of unknown nodes, their time complexity is still the same, all of which are $O(M \times (S - M))$. In the final stage of optimizing the location of unknown nodes, the time complexities of *CWDV-Hop* and *HWDV-Hop* are $O(I_1 \times P_1 \times (S - M))$ and $O(I_2 \times P_2 \times (S - M))$, respectively, while the time complexities of GDV-Hop and TWGDDV-Hop are $O(I_3 * P_3 \times (S - M))$. Generate additional time complexity $O(I_3 * P_3)$ to compute the fitness function, and require $O(I_3 \times)$ to update the position of the gray wolf. Thus, the time complexity of *TWGDV-Hop* has somewhat grown since the use of improved *GWO* to optimize the node positions in *WSN*. Sustained by current high-performance data processing platforms, the somewhat grown time consumption of the algorithm in this paper can be omitted, but it significantly improves localization accuracy.

## 6. Conclusions

The location in WSN is a very important topic. To increase the localization accuracy of *DV-Hop*, this paper proposed *TWGDV-Hop*. Considering the error caused by varying hop lengths, three communication radius broadcast messages were adopted, and a hop difference correction coefficient was introduced to further correct the minimum hops. In light of the influence of distance on different anchor points, *TGWDV-Hop* adopts a square quasi-measurement strategy to calculate *ADPH* of anchor nodes, and introduces distance weighted hop distance to jointly correct *ADPH*. On this basis, an improved *GWO* is adopted to reduce the error in estimating coordinates. Although the time complexity during the optimization operation increases slightly, it is acceptable. The experimental results display that *TGWDV-Hop* can minimize node positioning errors to the maximum

extent. Expanding the proposed scheme to estimate the position of unknown 3D sensor nodes is our future research direction.

**Author Contributions:** Conceptualization, X.Y. and W.Z.; methodology, X.Y. and T.L.; software, W.Z. and C.T.; validation, X.Y. and W.Z.; investigation, X.Y. and W.Z.; writing—original draft preparation, X.Y. and W.Z.; writing—review and editing, X.Y. and W.Z.; supervision, X.Y. and C.T.; project administration, W.Z. All authors have read and agreed to the published version of the manuscript.

**Funding:** This research was funded by the Open Project of Anhui Key Laboratory of Intelligent Building and Building Energy Efficiency of Anhui University of Architecture and Architecture (IBES2020KF03), Anhui Provincial Key Research and Development Program (202104g01020005), State Key Laboratory of Tea Biology and Resource Utilization of Anhui Agricultural University (SKLTOF20220131), Domestic Visiting Program for Outstanding Young Teachers in Colleges and Universities (gxgnfx2021154,gxgnfx2022078), Natural Research Science Institute of Anhui Universities (2022AH051372), Subsidy projects for surplus funds of Suzhou University (szxy2023jyjf86, szxy2023jyjf80, szxy2023jyjf71).

**Data Availability Statement:** Not applicable.

**Conflicts of Interest:** The authors declare no conflict of interest.

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
