# Peer review of "A Novel Localization Technology Based on DV-Hop for Future Internet of Things"

_electronics, doi:10.3390/electronics12153220_

Round 1

Reviewer 2 Report

This manuscript presented an improved algorithm, TWGDV-Hop aimed at enhancing the localization accuracy of DV-Hop in WSN. The technique addresses the error due to varying hop lengths by utilizing three communication radii, a hop difference correction coefficient, a square quasi-measurement strategy, distance weighted hop distance, and an improved Grey Wolf Optimization algorithm to minimize estimation errors in node coordinates. Experiments were conducted and results were critically analyzed and presented.

The paper is interesting, scholarly written, and technically and logically sound. The methodology was comprehensively discussed and the methods employed were appropriate. The results were presented and logical conclusions were drawn.

 I, therefore, accepted the paper in its present form.

Author Response

Thank you very much for your recognition of our article.

Reviewer 3 Report

1- the authors present a novel approach for localization technology based on an improved DV-Hop algorithm. Then the enhanced ??? is used to evolve the place of each node to be located.

1- the title is too general, no specification is given, similarly the key word list  should be extended

2- the contribution is clearly presented 

3-the language needs deep proofreading

4- in the experimentation the authors presented the Localization error for different shapes, what about considering a random shape

Edition issues:

- references 1 to 6 are not in the suitable format  in line  20 and line 414

-the paragraph numeration should start from 1 and not 0, so the introduction is the part number 1

- paragraphs 2 and 2.1 have the same title??

- lines 208  to 214, the scientific notations should be rewritten using the equation editor, the same issue is present is many other mathematical notations and variables and equations. Please check all variables and equations

- figures should be centered

-  avoid using infinitif or imperative form in experimentation " line 444  445 for example

- table 2 is splitter in two pages

Need proofreading

Reviewer 4 Report

electronics-2490921

This article needs several improvements to properly reach the publication. Some of them are listed below:

1.      There are several grammatical mistakes and typos that must be corrected with detailed proofread.

2.      Additionally, some references are wrongly cited. Some references are in superscript i.e., [3][4][5][6] and some are not, i.e., [7], and so on. Keep consistency throughout the paper.

3.      Most of the abbreviations are repeated, such as Internet of Things, and so on. Please use the full form for the first time and then use the abbreviations throughout.

4.      Further increase the resolution of figures 1, 2 and other as well.

5.      The related work section needs to refined as the authors have a big paragraph which is too confusing and boring to read and understand. Please add a brief and concise paragraph.

6.      To know more about localization, the authors can refer to “A Review of Underwater Localization Techniques, Algorithms and Challenges” and “Extended Kalman Filter-based Localization Algorithm by Edge Computing in Wireless Sensor Networks.”

7.      For numerical equations, the authors have sometimes used the name “formula” and sometimes “equation”, please keep consistency throughout.

8.      The authors have added a series of numerical equations, but some of them are missing proper explanation and almost all equations are missing proper references citations.

9.      Figure 13 is completely blur and unable to read. Similarly, figures 14, 15, and so on.

10.   All the figures of this paper need to be carefully modified. Almost all figures are invisible.

11.   What is the difference between figures 13, 14, and 15??? The authors have just changed the number anchor nodes, else nothing. Instead, all these figures can be represented in one figure.

12.   At the end of the paper, there are two sets of references list? Why this?

13.   The key contribution of this work is very limited. Localization for IoT is a very general task nowadays. The authors should do some serious experiments to add proper results.

14.   The technical depth of this paper is not adequate too. The flow is missing. 

Extensive editing of English language required

Reviewer 5 Report

1. While the paper talks about the Internet of Things, it is more about the wireless sensor networks and improvement of their standard localization algorithm. As such the title and the article do not match well.

2. The flow of content in the paper can be significantly improved.

3. Some acronyms have neither been expanded nor briefly explained.

4. Figures lack clarity. Too many of them have been crammed in too little space making it difficult for the reader to be able to understand and be convinced of the results and conclusions.

While the overall quality and usage of the English language is okay, the article needs a few thorough reads because there are errors and a few things that cause confusion.

Round 2

Reviewer 1 Report

Authors have improved their paper which can now be accepted for publication.

Author Response

Thank you very much for your feedback on our article.

Reviewer 4 Report

The authors have addressed some of the comments but still I have some suggestions that must be considered. 

1.      English grammar should be improved.

2.      Add further details to figures captions. Especially, in figures 13-15 explain each part briefly in the caption.

3.      The authors mentioned “The localization errors of all above algorithms are manifested in tables 4-6”. Although, there are only 4 tables in paper. This is the carelessness of the author.

4.      I strictly recommend the paper to be revised carefully. Additionally, avoid the monotonous words from the paper. Such as the tables 2-4 captions, and so on.

Extensive editing of English language required. Moreover, there are several typos and errors. 
